# Structure of the class C orphan GPCR GPR158 in complex with RGS7-Gβ5

Eunyoung Jeong [1,2], Yoojoong Kim [1,2], Jihong Jeong [1] & Yunje Cho [1✉]

GPR158, a class C orphan GPCR, functions in cognition, stress-induced mood control, and synaptic development. Among class C GPCRs, GPR158 is unique as it lacks a Venus flytrap-fold ligand-binding domain and terminates Gαi/o protein signaling through the RGS7-Gβ5 heterodimer. Here, we report the cryo-EM structures of GPR158 alone and in complex with one or two RGS7-Gβ5 heterodimers. GPR158 dimerizes through Per-Arnt-Sim-fold extracellular and transmembrane (TM) domains connected by an epidermal growth factor-like linker. The TM domain (TMD) reflects both inactive and active states of other class C GPCRs: a compact intracellular TMD, conformations of the two intracellular loops (ICLs) and the TMD interface formed by TM4/5. The ICL2, ICL3, TM3, and first helix of the cytoplasmic coiled-coil provide a platform for the DHEX domain of one RGS7 and the second helix recruits another RGS7. The unique features of the RGS7-binding site underlie the selectivity of GPR158 for RGS7.

---

[1] Department of Life Science, Pohang University of Science and Technology, Pohang, Republic of Korea. [2] These authors contributed equally: Eunyoung Jeong, Yoojoong Kim. ✉email: yunje@postech.ac.kr

GPR158 is a class C orphan G protein-coupled receptor (GPCR) prominently expressed in brain tissue[1]. Although the pathological function of GPR158 is not clearly understood, the receptor is believed to play important roles in memory, stress-related mood control, and synaptogenesis[2–4]. In the CA3 region of the hippocampus of mice, GPR158 transduces the osteocalcin (OCN) signal and enhances memory, in part through inositol 1,4,5-trisphosphate (IP3) and brain-derived neurotrophic factor[2]. Increased levels of GPR158 have been observed in major depressive disorder patients and chronic stressed mice, whereas ablation of GPR158 in mice produced antidepressant-like affects[3]. GPR158 forms a trans-synaptic complex with proteoglycans of the extracellular matrix and controls the presynaptic differentiation of mossy fiber-CA3 synapses[4]. In addition, GPR158 is implicated in the development of prostate cancer[5].

The class C GPCR family comprises $Ca^{2+}$-sensing (CaSR), metabotropic glutamate receptors (mGluR1-8), γ-aminobutyric acid receptor B (GABA$_B$), and sweet taste receptors, which are characterized by a large extracellular domain and dimerization[6]. Two orphan GPCRs, GPR158 and GPR179, are the least characterized class C members, and share over 70% sequence similarity in both extracellular and TM domains[1]. Unlike other class C GPCRs, GPR158 and GPR179 devoid of a Venus flytrap (VFT)-fold in their extracellular domain, and transmit signals via noncanonical mechanism by which both GPCRs recruit the regulator of G protein signaling 7 (RGS7)–Gβ5 heterodimer to the plasma membrane[1,7,8]. Activation of the G protein by GPR158 remains unclear: although OCN binds to a complex containing GPR158 and Gαq and regulates the IP3 production, no functional assay using OCN as an agonist has been reported[2]. By contrast, GPR158 exhibits constitutive activity for Gi/o proteins but not for Gq[9]. The noncanonical signaling for GPR158 is more clearly established: GPR158 localizes RGS7–Gβ5 and the Gαi/o protein activated by other GPCRs, and allosterically promotes GTPase activity of Gαi/o, which ultimately reduces the activity of adenylate cyclase and controls other signaling pathways[7,8,10,11]. Because signal termination is a sensitive biological event, a GPCR must recognize RGS proteins with high specificity[12–18]. However, despite recent progress in GPCR structural biology, it remains unresolved how a GPCR specifically binds to RGS proteins at the molecular level[19]. Here, we report the structures of GPR158 alone (3.5 Å) and in complex with one or two RGS7–Gβ5 heterodimers (4.3, 4.7 Å), and provide insights into the noncanonical signaling mechanism by which the orphan GPCRs selectively recruit the RGS7–Gβ5 complex and regulate the signals.

## Results

**Overall structure**. We determined the overall structure of apo GPR158 at an average resolution of 3.5 Å (Supplementary Figs. 1 and 2 and Supplementary Table 1). GPR158 consists of an α/β-fold extracellular domain, a cysteine-rich (CR) domain with an epidermal growth factor (EGF)-like fold, and 7TM domain. GPR158 forms an elongated shape ($79 \times 147 \times 49$ Å$^3$) and dimerizes through both extracellular and TM domains (Fig. 1). Each domain of a GPR158 protomer is nearly identical to the corresponding part of another protomer with root-mean-square deviation (r.m.s.d.) ranging from 0.5 to 1.1 Å. However, entire GPR158 protomers differ significantly with an r.m.s.d. of 3.3 Å, indicating the flexibility between the domains within a protomer (Supplementary Fig. 1h). Overall, the GPR158 structure adopts a unique topology in extracellular and linker domains as well as its TM interface arrangement, different from other class C GPCRs[6].

**Structure of the extracellular domain**. The extracellular domain consists of the N-terminal three helices and the core. The core is comprised of a six-stranded sheet at the center, two vertically aligned helices on one face, and a horizontally aligned helix on another face (Fig. 2a, b). A DALI search revealed that the core structure most resembles the Per-Arnt-Sim (PAS) domain that is frequently observed in signaling proteins related to circadian, light and oxygen sensing[20,21]. Structural comparison with various PAS domains showed that the extracellular core of GPR158 has the r.m.s.d. values of 2.1–2.5 Å for 145 Cα atoms with transducer-like protein 3 (Tlp3), histidine kinase (HK)-Z3, and HK-Z6 (refs. [22,23]) (Supplementary Fig. 3).

The extracellular domain dimerizes through the side-by-side packing of α4 and α5′ and α5 and α4′ helices (′ indicates the second protomer) in a parallel manner, with a buried surface area of 1806 Å$^2$ (Fig. 2a, c and Supplementary Fig. 4a). At the center of the interface, a hydrophobic cluster formed from W156′ and L160′ (α5′) and M139 and F135 (α4) is packed against a symmetrical hydrophobic cluster, and the resulting hydrophobic network stabilizes the extracellular dimer. Below the PAS domain, the N-terminal three antiparallel helices are placed at the midpoint while forming close contacts with both PAS and CR domains (Figs. 1c and 2a). The three helices do not interact with the PAS core. Instead, helix α2 interacts with the β4–β5 loop and a B loop of the CR linker. The CR linker consists of two parts: the top two-thirds (A-C loops) folds into an EGF-like structure and the bottom third (D loop) forms a spiral-loop structure (Fig. 2d). The EGF-like and spiral motifs are stabilized through three and two disulfide bonds, respectively, which are also conserved in GPR179 (Fig. 2d and Supplementary Figs. 4b and 5a). The EGF-like linker can be superimposed to other EGF-like domains with r.m.s.d. values of 2.0–2.3 Å for 66 Cα atoms[24–27] (Supplementary Fig. 5b, c). A pairwise comparison with the Factor IX EGF-like domain reveals that the conserved D314 and D316 (A loop) and the main chain oxygen atoms of the elongated B loop correspond to the residues that coordinate a $Ca^{2+}$ ion[24] (Supplementary Fig. 5a, c). The D loop interacts with ECL2, which is connected to TM3 via the C481$^{3.29}$-C573$^{ECL2}$ disulfide bond (numberings of class A and class C GPCRs are based on BW[28] and Pin[29], respectively; Fig. 3a and Supplementary Fig. 5d).

In another subclass, we observed an apo GPR158 structure in which the CR linker is significantly compressed to 20 from 47 Å (Supplementary Figs. 1g and 5e). Because the resolution of this region is not sufficient to build a model for this subclass structure, it is unclear which part is responsible for the reduced length of the CR linker. Conformational change of the GPR158 CR linker suggests that the region between the PAS and TM domains undergoes dynamic movement, implicating a role for this linker in communication between the two domains.

**Comparison of the GPR158 TM domain with other class C GPCRs**. The TM domains of the two protomers are virtually identical with an r.m.s.d. of 0.5 Å. The GPR158 TM domain is most similar to inactive GBR2 (1.8 Å) and most distant from active mGluR5 (3.1 Å) in class C GPCRs[30–37] (Supplementary Fig. 6a). Compared with other class C GPCRs, GPR158 TM4 is longer by one and a half turns at the extracellular end (Supplementary Fig. 6b). The most notable feature of a TM domain is extensive ionic networks present in three layers (Fig. 3a, b). While ion pairs in layers I and II are unique in GPR158, the ionic lock in layer III is conserved in other GPCRs. Layer I located below ECL2 contains an R485$^{3.33}$-D579$^{5.40}$-R631$^{6.57}$ network, to which Y647$^{7.32}$ makes an H-bond contact (Fig. 3b). Beneath this layer, E586$^{5.47}$ engages in ionic contact with R488$^{3.36}$ and H628$^{6.54}$, which are buttressed by a polar network formed by Y496$^{3.44}$,

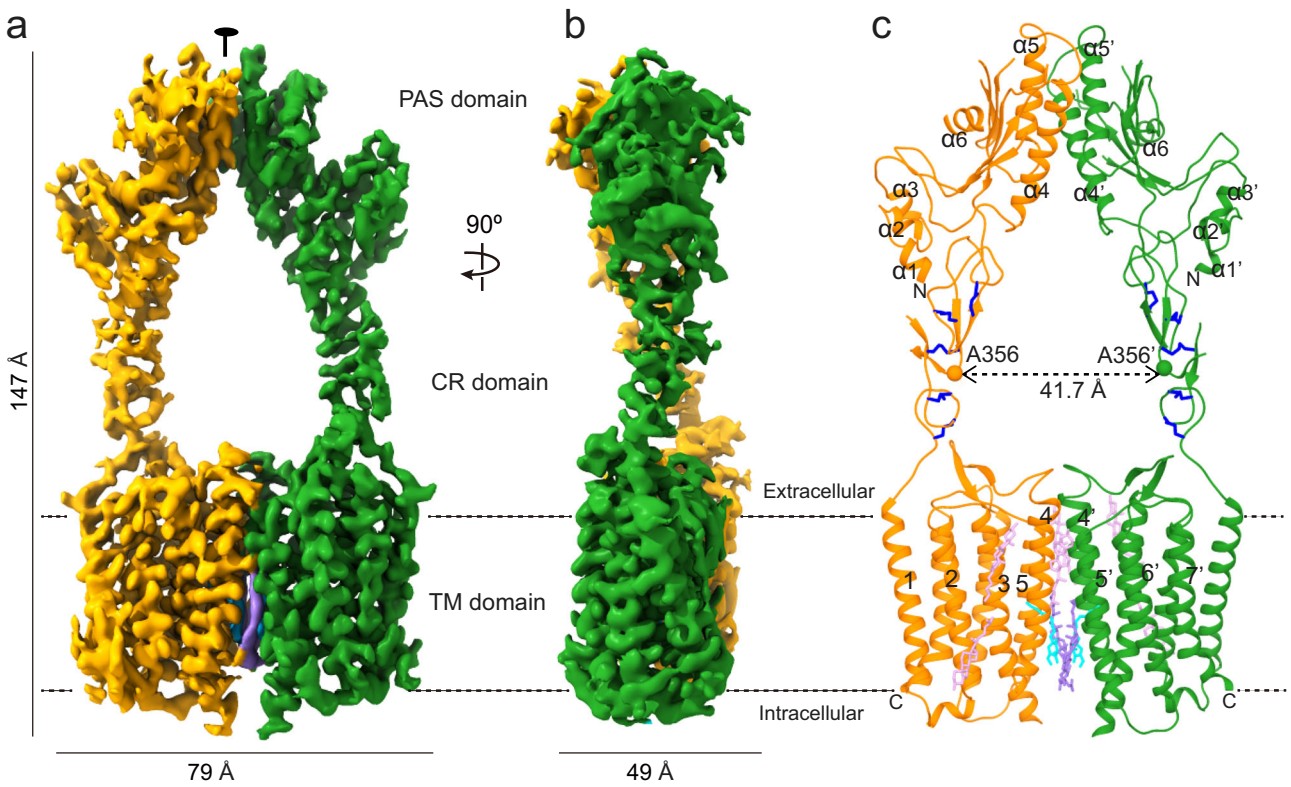

**Fig. 1 Cryo-EM structure of GPR158. a, b** Cryo-EM maps of the GPR158 homodimer in orthogonal views. **c** Model of GPR158 colored the same as in **a**. CHS, OG, and cholesterol molecules are represented as purple, sky blue, and pink sticks, respectively. Disulfide bonds are shown as blue sticks.

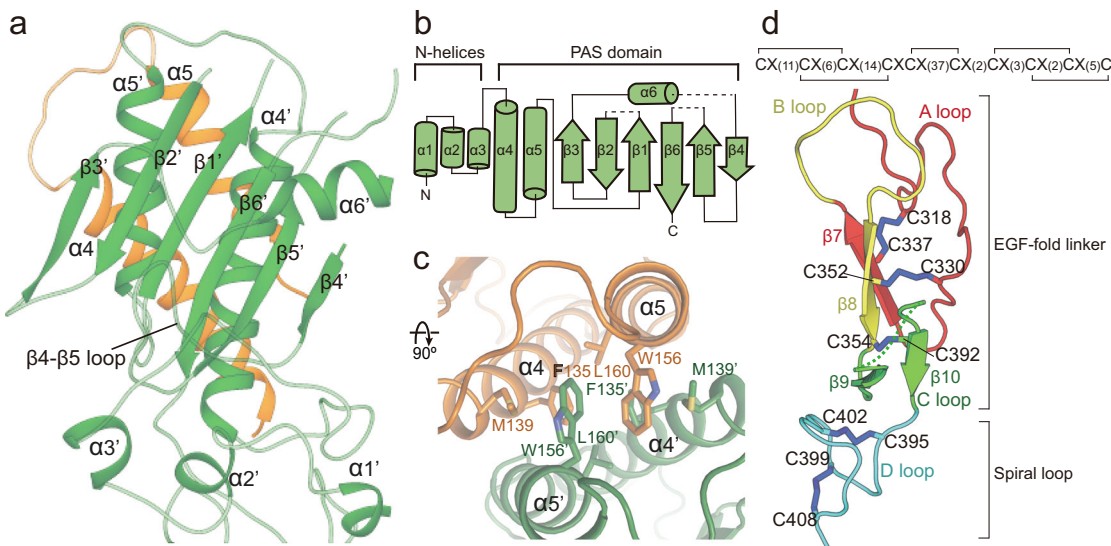

**Fig. 2 Structure of the extracellular domain of GPR158. a** Structure of the N-terminal helices (bottom) and the PAS domain (top) of GPR158 in the view shown in Fig. 1b. **b** Topology diagram of the N-terminal helices and the PAS core of GPR158. Dotted lines indicate disordered loops. **c** Close-up view of the dimeric interface between the GPR158 PAS domains in a top view from **a**. **d** The CR domain stabilized by disulfide bonds shown in black lines (top) and blue sticks (bottom).

E620[6.46], S624[6.50], H650[7.35], and T654[7.37]. Layer III contains an ionic lock, a hallmark of the inactive state in other GPCRs, by tethering TM3, TM6, and ICL3. In this ionic network, the K666[7.51]-E609[ICL3]-K502[3.50] cluster is positioned near the cytoplasmic end. K502[3.50] also forms an H-bond with S450[ICL1]. K666[7.51] from the LxPKxx motif is equivalent to FxPKxx in TM7 in other class C GPCRs and NPxxY in class A GPCRs[31]

(Supplementary Fig. 7a). K502[3.50] is equivalent to K[3.50] that interacts with D688 in the D/ERY motif in ICL3 in the GABA$_B$[31–35]. Overall, the extensive ionic and polar networks in the three layers stabilize TM3, TM5, TM6, TM7, ICL1, and ICL3 in apo GPR158 (Fig. 3c). The polar networks of the top two layers occupy the orthosteric or allosteric binding sites in other GPCRs[38–43] (Fig. 3d, e). As a result, the space inside the GPR158

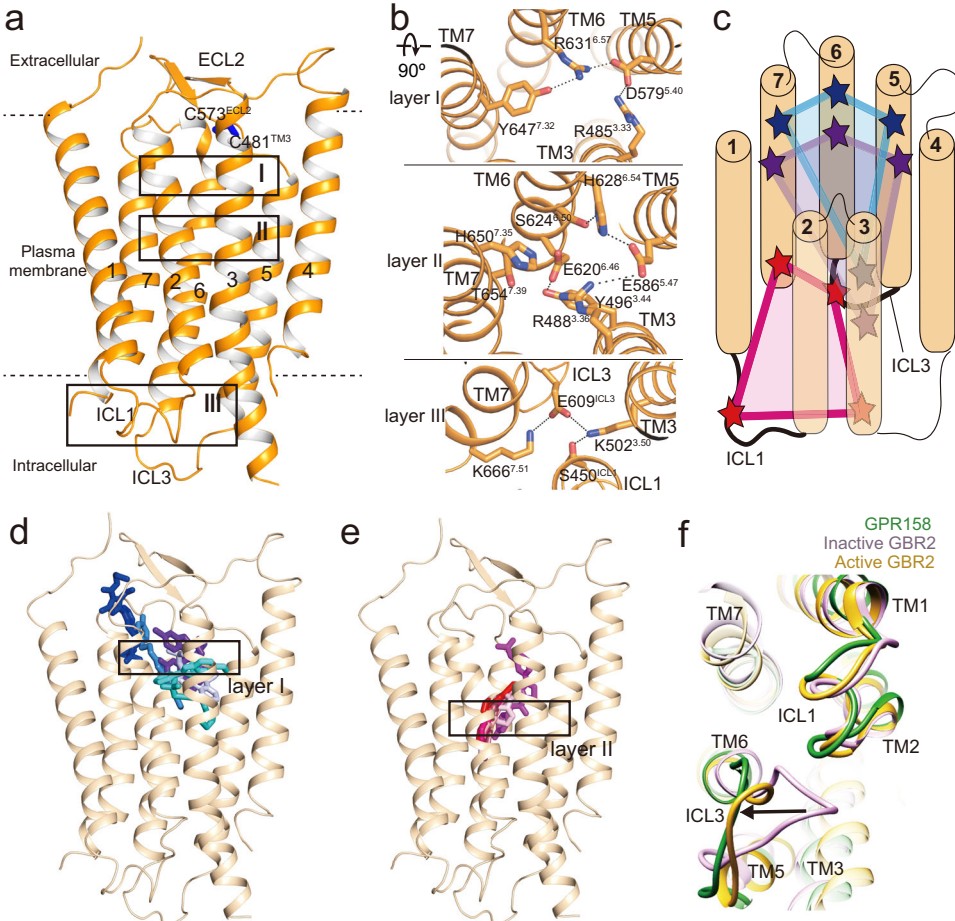

**Fig. 3 Structure of the TM domain in comparison with other GPCRs. a** Overall structure of the 7TM domain of GPR158. The opposite protomer is omitted for clarity. Boxes include clusters of interactions within a TM domain. The disulfide bond between ECL2 and TM3 are colored blue. **b** Close-up views of the three layers of ion pairs within a TM domain in an orthogonal view from **a**. The ionic and polar interaction network in layer I (top) and II (middle) stabilize TM3, 5, 6, and 7. **c** Cartoon representation of intra-TM domain interactions formed by layer I (blue), II (purple), and III (pink). **d**, **e** Location of the orthosteric ligands bound to class A (**d**) and the allosteric ligands bound to class C (**e**) GPCRs in comparison with layer I and II within the GPR158 TM domain (light orange). The ligands are shown as sticks and colored as follows: GPR52 (6LI0, blue), MT1 (6ME3, cyan), mAChR (6OIJ, purple blue), A2A (6SOL, sky blue), OX1R (6V9S, purple), and ADRB2 (6OBA, light blue) from class A (**d**), and mGluR1 (4OR2, magenta) and mGluR5 (4OO9, light pink; 5CGC, red; 5CGD, raspberry; 6FFH, salmon; 6FFI, hot pick) from class C (**e**). **f** Comparison of TM domain structures between GPR158 and inactive or active GABA_B by aligning TM domains of GPR158 with the corresponding GBR2 structures. Conformational differences in ICLs between GPR158 and inactive GBR2 are indicated by an arrow.

TM domain is not sufficient for the ligand binding. Two cavities within the GPR158 TM domain are estimated to be 286 and 464 Å$^3$, significantly smaller than those in other class C or class A GPCRs, which range from 833 to 2832 Å$^3$[38–43] (Supplementary Fig. 6c).

Another notable feature of the GPR158 TM domain is the ICL1 and ICL3 conformation. In GPR158, ICL1 and ICL3 are straight and directed away from TM3 and TM6, resulting in open conformations similar to those in the active state of GABA_B[32–34] (Fig. 3f). Thus, the structure of the GPR158 TM domains exhibit structural features of both inactive and active states of class C GPCRs.

**Dimeric arrangement of the TM domains.** In apo GPR158, the TM4/5 helices are arranged in an inverted V-shape at the dimeric interface (Figs. 1c and 4a). The residues at the top half of TM4 and TM5 near the extracellular end make direct contacts each other, whereas the bottom half at the intracellular end interact only through lipid molecules (Fig. 4a). The top half of the

interface buries 1623 Å$^2$ of the surface area. Near the extracellular end, W539[4.45] and F540[4.46] are packed against W578[5.39], and the hydrophobic cluster interacts with M581′[5.42], W578′[5.39], and F540′[4.46] of an opposite protomer (Fig. 4b). The other side of W578[5.39] packs against W539′[4.45] and buttresses the interactions described above. W539[4.45] is replaced by Gly in GPR179, whereas W578[5.39] and F540[4.46] are conserved (Supplementary Fig. 4c). Assuming that GPR179 forms a similar dimer, the structure suggests that the W578[5.39]–F540[4.46] contact is critical for the formation of a dimeric interface (Fig. 4b). In GBR1, GBR2, and mGluR5, F540[4.46] is replaced by Thr or Ile, whereas W578[5.39] is conserved or replaced by Gly, explaining why the formation of the TM4/5 interface is favorable in GPR158 (Supplementary Fig. 7a). At the bottom half of the TM interface, we observed well-defined but unknown densities and assigned them as putative octyl-β-D-glucopyranoside (OG) and cholesterol hemisuccinate (CHS), which were added during sample preparation (Fig. 4a, b and Supplementary Fig. 2g).

The inverted V-shaped arrangement of the TM4/5 helices at the GPR158 interface is in marked contrast to other class C

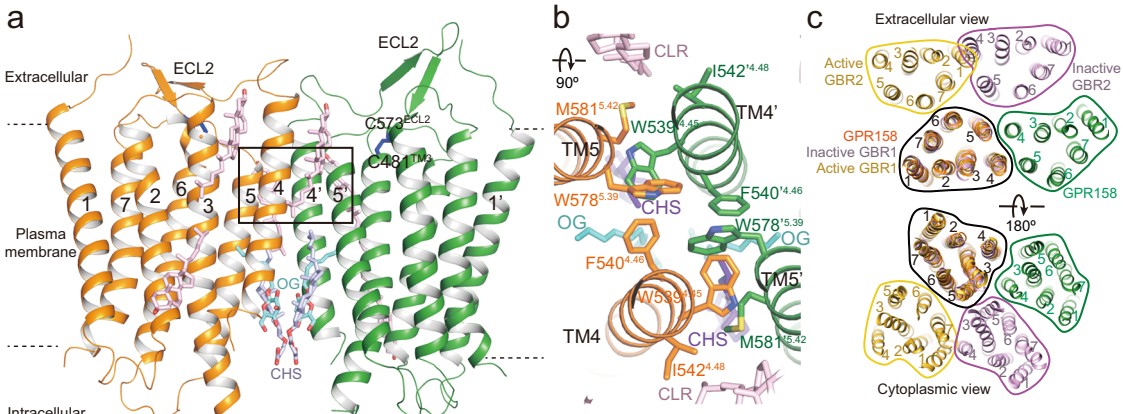

**Fig. 4 Dimeric arrangement of the TM domains. a** Overall structure of the 7TM domains of the GPR158 homodimer. CHS, OG, and cholesterol molecules are represented as purple, sky blue, and pink sticks, respectively. A box includes clusters of interactions at the dimeric interface of the TM domains. The disulfide bonds between ECL2 and TM3 are colored blue. **b** The TM4/5 dimeric interface looking down from top of **a**. **c** Comparison of the dimeric interface of GPR158 7TM domains with inactive and active GABA$_B$ receptors. The 7TM domains of inactive and active GBR1 are superposed to one GPR158 protomer (black line). Both extracellular (top) and cytoplasmic (bottom) views are shown.

GPCRs[30–37]. Upon superposition of the first TM domains of class C GPCRs, the second TM domain of GPR158 is rotated by 90° with respect to the second TM domain of inactive GABA$_B$ and mGluR5 in a counterclockwise direction, and rotated by 60° relative to that of CaSR (Fig. 4c and Supplementary Fig. 6d). Similar to apo GPR158, TM helices at the interface in active GABA$_B$ form an inverted V-shape. However, TM6/7 rather than TM4/5 are arranged at the dimeric interface. Because all class C GPCRs involve TM6–TM6 interface formation in the active state, we hypothesized that GPR158 might adopt a similar conformation upon activation; hence, we modeled a putative GPR158 dimer by aligning each GPR158 TM domain onto GBR1 and GBR2 of active GABA$_B$ (Supplementary Fig. 6e). Although several hydrophobic residues pack against each other, positively charged residues (Arg637 at the extracellular side; Arg601, H608, and Arg611 at the cytoplasmic side) face each other at the interface, resulting in repulsion. This suggests that the GPR158 TM domains are unlikely to form a TM6–TM6′ interface in their current conformations.

**Structures of the GPR158–RGS7–Gβ5 complex.** The GPR158 TM domain exhibits both inactive and active features of class C GPCRs; hence, we cannot elucidate whether apo GPR158 is in an inactive, partially active or fully active state that can directly couple to the Gαβγ heterotrimer. However, this unusual feature might be explained by the noncanonical signaling mechanism of GPR158, in which the apo GPR158 receptor recruits and inactivates the Gαi/o protein via the RGS7–Gβ5 heterodimer[7,8]. The apo GPR158 recruits the RGS7–Gβ5 heterodimer and Gαi/o protein via the first- and second half of the cytoplasmic domain, respectively[7]. Cryo-EM analyses for GPR158 (residues 1–863)–RGS7–Gβ5 complex revealed three types of GPR158 states: apo GPR158 and the GPR158:RGS7:Gβ5 complex in a 2:1:1 ratio at an average resolution of 4.3 Å and in a 2:2:2 ratio at 4.7 Å (Fig. 5a, Supplementary Figs. 8 and 9 and Supplementary Table 1). In both complexes, structures of GPR158 and the four domains of RGS7 DEP (Dishevelled, Egl10, Pleckstrin), DHEX (DEP helical extension), GGL linker, and the RGS domain, and the Gβ5 subunit were clearly visible and their models fitted well into the cryo-EM maps (Supplementary Fig. 9o, p). However, most of the cytoplasmic domain of GPR158 (residues 669 to 863) is disordered except for a coiled-coil (Ha and Hb) comprising 61 residues (Supplementary Figs. 4d and 9s).

In this coiled-coil, the first helix (Ha) forms seven turns from 27 residues, and the second helix (Hb) of 22 residues forms six turns, and they are connected by an eight-residue linker (Supplementary Figs. 4d and 9s). Although we could not assign the side chains of the cytoplasmic helices, biochemical studies and secondary structure prediction suggest that this region is likely to correspond to residues 708 to 763 (refs. [7,8]). The two helices are rich in Leu and Ile, suggesting that they form a coiled-coil through these residues, consistent with the three-dimensional (3D) model predicted using the I-TASSER program[44] (Supplementary Fig. 9t).

Since the 2GPR158–RGS7–Gβ5 and 2GPR158–2RGS7–2Gβ5 complexes exhibit virtually identical structures, except for an additional RGS7–Gβ5 heterodimer in the latter, we focus on describing the overall structure of the 2GPR158-2RGS7-2Gβ5 complex and close-up view of the 2GPR158–RGS7–Gβ5 complex (Fig. 5a and Supplementary Fig. 10a). The RGS7–Gβ5 heterodimer binds to GPR158 through the DHEX domain, which is sandwiched between the ICLs and the Ha helix (Fig. 5a and Supplementary Fig. 10a, b). The disordered ICL2 (10 residues) and the cytoplasmic tip of TM3 in apo GPR158 becomes ordered upon binding of RGS7 (Fig. 5b and Supplementary Fig. 9q). Conformation of the three ICLs of GPR158 bound to the RGS7 are similar to those of Gi-bound GABA$_B$, mGluR2, and mGluR4[34,36,45,46] (Fig. 5c). The RGS7–Gβ5 complex is tilted ~40° relative to the plasma membrane in a way that both the DEP and DHEX domains are located closest to the TM domains, and the RGS domain is farthest (Fig. 5a and Supplementary Fig. 10a). The DHEX domain faces the ICLs, inserting the Xα1 helix and the Xα1Xα2 loop to the inverted V-shaped TM interface (Fig. 5b). This loop is also involved in binding to R7BP[47]. The Xα1 and Xα3 helices of the DHEX domain are packed below the GPR158 TM (ICL2, ICL3, and TM3), and the Xα3 helix is on top of the Ha helix that is positioned in perpendicular to the TM helices (Fig. 5a, d and Supplementary Fig. 10a, b, c). The linker between the end of TM7 and the Ha helix is disordered and the distance between the two ends of the disordered region is 47 Å in one protomer and 53 Å in the other. Thus, it is unclear to which protomer the coiled-coil belongs. The GPR158 fragment (residues 665–775) containing the coiled-coil is sufficient for binding to the RGS7–Gβ5 complex, illustrating the importance of the Ha helix in binding to RGS7 (ref. [7]). Superposition of the RGS7–Gβ5 dimer in the complex onto the reported structure of free

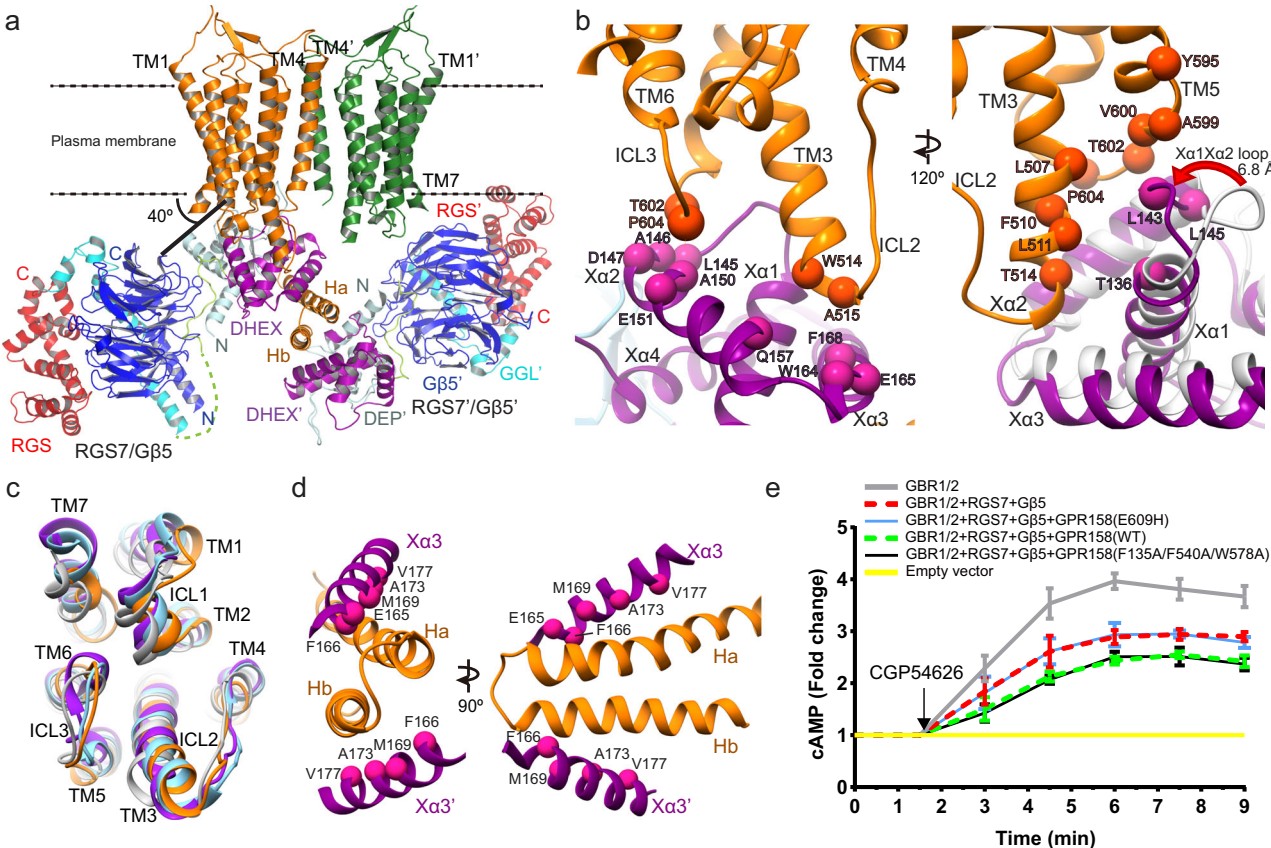

**Fig. 5 Cryo-EM structures of the GPR158−RGS7−Gβ5 complex. a** Overall structure of the 2GPR158−2RGS7−2Gβ5 complex. Each domain is colored as follows: each 7TM domain of GPR158, orange and green, respectively; DEP, light cyan; DHEX, purple; GGL-RGS linker, light green; GGL, cyan; RGS, red; Gβ5, blue. The disordered region between DHEX and GGL domains is indicated by a light green dotted line. **b** Close-up view of the interaction between the DHEX domain and GPR158. Residues at the interface are shown as spheres. The 120°-rotated view (right) highlights the conformation change of the Xα1Xα2 loop upon GPR158 binding. The RGS7 bound to GPR158 is aligned with the RGS7−Gβ5 dimer[47] (PDB 6N9G, light gray). **c** Superimposed 7TM domain of GPR158 bound to RGS7−Gβ5 (orange) with each 7TM domain of the GABA$_B$-Gi (PDB 7EB2, light gray), mGluR2-Gi (PDB 7MTS, purple), and mGluR4-Gi (PDB 7E9H, sky blue) viewed from the bottom. **d** Interactions between Xα3 and the cytoplasmic helices in two views. **e** CGP54626-induced cAMP production using HEK293T cells expressing indicated GPR158 mutants. Data represent mean values with standard deviations from three independent experiments.

RGS7−Gβ5 showed that the two structures are almost identical with an r.m.s.d. of 1.3 Å[47]. However, the Xα1Xα2 loop of DHEX is shifted towards TM5 by as much as 6.8 Å, consistent with previous reports that the loop is the most flexible part of the RGS7 protein[47] (Fig. 5b).

To understand if the TM domain arrangement of the GPR158 dimer is important for binding of RGS7−Gβ5, we aligned a TM domain of the GPR158−RGS7−Gβ5 complex to a TM domain of inactive or active CaSR, mGluR5, and GABA$_B$ receptors[30−35](Supplementary Fig. 11a, b). In all cases, RGS7−Gβ5 collides to TM domain of the other protomer. This suggests that the inverted V-shaped TM4/5 arrangement of apo GPR158 is critical for recruiting RGS7−Gβ5, and explains why apo GPR158 can facilitate noncanonical signaling.

Another RGS7−Gβ5 heterodimer binds to the Hb helix, which is visible only in one GPR158 protomer (Fig. 5a, d and Supplementary Fig. 10d). The second RGS7−Gβ5 dimer is oriented in pseudo-twofold symmetry with respect to the first dimer against the Ha helical axes (Fig. 5a). Similar to the first dimer, binding of the second RGS7−Gβ5 complex is achieved through helix Xα3′ from the DHEX′ domain (Fig. 5d and Supplementary Fig. 10d). While DHEX′ of the second RGS7 binds to the Hb helix, the RGS′ domain of RGS7′ shifts proximal

to the micelles (Supplementary Movie 1). Thus, although the interaction between the second RGS7 and Hb helix is not as extensive as the interaction of the first RGS7, the RGS7′-micelle interaction contributes to the stability of the second RGS7−Gβ5 in the GPR158−RGS7 complex.

Apo GPR158 not only provides a platform for the binding of RGS7 and Gαi/o activated by other GPCRs such as GABA$_B$, but also potentiates GTP hydrolysis, thereby terminates both Gα and Gβγ signaling pathways[7,8,11]. To examine if the GPR158 structure affects the cAMP production, we engineered two different classes of GPR158 mutants in which the dimeric interface or intra-TM ion pairs were disrupted, and examined their cAMP-producing activity (Fig. 5e and Supplementary Fig. 12). Overexpression of the RGS7−Gβ5 complex alone decreased the amount of cAMP produced by adenylate cyclase as previously reported[11]. Co-expression of GPR158 and RGS7−Gβ5 further reduced the cAMP production. For the first class mutant, we simultaneously replaced the residues at the extracellular (F135A) and TM (F540A, W578A) domains (Figs. 2c and 4b). The dimer-disrupting mutant exhibited similar cAMP-producing activity relative to wild-type GPR158 (Fig. 5e and Supplementary Fig. 12a). Next, we mutated the intra-TM ion pairs in layer III (E609H) (Fig. 3b). Cells expressing the GPR158 E609H mutant exhibited increased cAMP

production compared with those expressing wild-type GPR158, similar to those lacking GPR158 (Fig. 5e and Supplementary Fig. 12a). This result suggests that a GPR158 protomer is sufficient for binding of the RGS7–Gβ5 heterodimer and Gαi/o protein and potentiating GTP hydrolysis activity by RGS7. However, TM domain structure is important for the activation of noncanonical signaling.

## Discussion

The PAS domain functions as a molecular sensor in various signaling pathways in all kingdoms of life[21,48]. Because of its general role in recognizing diverse ligands and proteins, the PAS domain has been observed in various signaling proteins including histidine kinases, nucleotide cyclases and response regulators[48]. In this work, we extend diversity of the PAS domain and present the first example of 7TM fused to the PAS domain in GPCRs. Because GPR158 and GPR179 are homologous, GPR179 is expected to adopt a similar overall structure. Structural similarity of the GPR158 extracellular core with those of chemotaxis receptor and histidine kinases suggests that GPR158 might interact with ligands through a conserved binding-pocket[22,23] (Supplementary Fig 3a, c). Alternatively, GPR158 might interact with the PAS-associating proteins[49].

The TM domains of apo GPR158 adopt a conformation reflecting both active and inactive states of other class C GPCRs. Furthermore, the TM4/5 interface is a unique feature of apo GRP158. Although these features are complicated to explain the activation of G protein via agonist-dependent direct coupling, the structure provides a clue for the RGS7–Gβ5 binding. Open conformation of ICL3 that resembles active GBR2 contributes to the interaction with the DHEX domain of RGS7 (Figs. 3f and 5b). The RGS7–Gβ5 heterodimer cannot be localized to the TM domains in inactive or active state of other class C GPCRs, which suggests that only the TM dimer with the TM4/5 interface in a GPR158 geometry can recruit the RGS7 complex (Supplementary Fig. 11a, b). Although a GPR158 monomer can recruit the RGS7 complex and facilitate GTP hydrolysis, it is possible that the ligand binding further accelerates GTPase activity in GPR158 dimer as previously shown in melatonin MT$_1$ receptor[50]. Furthermore, dimerization may be important in canonical signaling, in which binding of an agonist to a dimer directly activates G protein.

Termination of signaling is a highly sensitive and critical event, which can be controlled by the specific interactions between RGS proteins and subsets of Gα proteins as well as between GPCRs and RGS proteins[12,51,52]. Although relatively few examples of GPCR and RGS interactions have been identified, several GPCRs directly and selectively recruit various RGS proteins[12]. In particular, among class C GPCRs, GPR158 is unique in that it only interacts with RGS7 and RGS6 in the RGS family and recruits them to the plasma membrane in order to regulate G protein activity[1]. The GPR158–RGS7–Gβ5 structure provide a basis for the selectivity of GPR158 for RGS proteins: first, only RGS7 family members contain the DHEX domain, which is responsible for binding GPR158. In many RGS proteins, the DEP domain has been reported to interact with GPCRs[53,54]. Second, the relative orientation of DEP and DHEX domains in RGS7 and RGS9 differs substantially[47,55]. Furthermore, the helices in the DHEX domain are organized differently in these RGS proteins[47]. Aligning RGS9 with RGS7 in the GPR158–RGS7-Gβ5 complex structure results in steric collision between RGS9 and GPR158 (Supplementary Fig. 10e). Third, the Xα1Xα2 loop, a key element in recognizing GPR158, differs significantly from the RGS9 loop (Supplementary Fig. S10e). Furthermore, RGS6 and RGS7 share highly conserved sequence of the Xa1Xa2 loop, whereas RGS9

and RGS11 exhibited clear differences (Supplementary Fig. 7b). The conformation of the Xα1Xα2 loop changes upon binding to GPR158 (Fig. 5b). Since this loop is also involved in R7BP binding[47], the Xα1Xα2 loop might be an important mediator of RGS7 in binding to other GPCRs. Fourth, the GPR158-contacting residues in the DHEX domain are highly conserved in RGS6, but not in RGS9 and RGS11 (Supplementary Fig. 7b). Conversely, the cytoplasmic coiled-coil of GPR158 is crucial for recruiting the DHEX domain of RGS7 (ref. [7]). The cytoplasmic coiled-coil and the DHEX-contacting residues are highly conserved in GPR179, but not in other GPCRs (Supplementary Fig. 4d). These data suggest that GPR158 and GPR179 bind RGS7 protein in a similar manner. Furthermore, because the RGS7-binding region of GPR158 shares similarity with R7BP and R9AP, it is possible that the binding mode of R7BP and R9AP proteins for RGS7 is similar to that of GPR158.

GPCRs are known to interact with various RGS proteins via their ICL3 or C-terminal tail[12]. The GPR158–RGS7 interaction is much more extensive than other reported GPCR-RGS interactions. The GPR158–RGS7–Gβ5 complex structures suggest that direct coupling of GPR158 with G protein via ICLs similar to those observed in GABA$_B$, mGluR2, mGluR4, and Ste2 is unlikely in the presence of RGS7–Gβ5 complex because the RGS7–Gβ5 complex and G protein would collide each other[36,45,46,56] (Supplementary Fig. 11c). Instead, GPR158 recruits the Gαi/o subunit in various states through the first half of the cytoplasmic domain and places the RGS7 and Gαi/o proteins in close proximity. Because the ligand-free GPR158 can stimulate GTPase activity, it would be reasonable to speculate that apo GPR158 triggers rearrangement of RGS7 and Gi proteins to interact each other. Further structural analysis is required to determine how binding of Gα to the cytoplasmic half activates RGS7. It is unclear if the apo GPR158's activity that potentiates GTP hydrolysis is partial or full. However, stimulation of GTP hydrolysis by the ligand-free GPR158 is critical because GPR158–RGS7–Gβ5 complex turns off the Gi/o signaling activated by other GPCRs and regulates ion channels such as G protein-coupled inwardly-rectifying potassium and voltage-gated calcium channels[10,11]. Finally, structural information of the GPR158 PAS domain and the GPR158–RGS7 interface can provide an attractive framework to design antidepressant drugs[3].

## Methods

**Expression and purification of apo GPR158.** The gene encoding human GPR158 was purchased from Addgene (#66332), amplified by PCR, and cloned into the pEG BacMam vector using restriction enzymes *Eco*RI and *Not*I[57]. The resulting GPR158 constructs encoding amino acid residues 1–710 or residues 1–863 were fused with a PreScission protease cleavage site, green fluorescent protein (GFP), and a Twin-Strep II tag at the C-terminal end. Baculoviruses harboring GPR158 were generated in *Spodoptera frugiperda* (Sf9) cells using the Bac-to-Bac system (Invitrogen). For expression of apo GPR158, HEK293S GnTI$^-$ cells grown to a density of $3.0 \times 10^6$ cells/ml were infected with recombinant baculoviruses and cultured for 12 h at 37 °C. Sodium butyrate (10 mM) was added, and cells were further incubated for 48 h at 30 °C.

Cells were collected by centrifugation and resuspended in lysis buffer containing 20 mM HEPES pH 7.5, 300 mM NaCl, and 1 mM EDTA. Cell membranes were solubilized with 20 mM HEPES pH 7.5, 300 mM NaCl, 1% (w/v) lauryl maltose neopentyl glycol (LMNG; Anatrace), 0.1% (w/v) CHS; Sigma), 20% glycerol, 1 mM EDTA, 1 mg/ml iodoacetamide, 150 μg/ml benzamidine, and protease inhibitor cocktail (Roche) for 1 h, and then cleared by ultracentrifugation at $125,171 \times g$ with a 45Ti rotor (Beckman) for 1 h at 4 °C. The supernatant was isolated and applied to StrepTactin XT resin (IBA Lifesciences) for 3 h. The resin was thoroughly washed with 20 mM HEPES pH 7.5, 300 mM NaCl, 0.1% LMNG, 0.01% CHS, 10% glycerol, and 1 mM EDTA. To elute the resin-bound protein, PreScission protease was added at a ratio of 10:1 (w/w) and incubated overnight in buffer comprising 20 mM HEPES pH 7.5, 200 mM NaCl, 0.01% LMNG, 0.001% CHS, 10% glycerol, and 5 mM EDTA. The resin was transferred to a gravity column, and the flow-through fraction containing GFP-cleaved GPR158 was collected. The protein was concentrated to 6.9 mg/ml using an Amicon Ultra centrifugal device (100 kDa cut-off; Millipore) and stored at −80 °C.

**Expression and purification of the GPR158–RGS7–Gβ5 complex.** Genes encoding human RGS7 (#55760) and Gβ5 (#55763) were purchased from Addgene and cloned into pEG BacMam vectors to include a Flag tag fused at the C-terminal end of RGS7. Expression of the GPR158–RGS7–Gβ5 complex was induced as described above for apo GPR158, except that three kinds of baculoviruses each harboring GPR158, RGS7, or Gβ5 were used at a ratio of 1:1:1. Cultured cells were collected by centrifugation at 4647 × g for 30 min, and lysed in buffer containing 20 mM HEPES pH 7.5, 500 mM NaCl, and 1 mM EDTA. Cells were disrupted on ice using a Dounce homogenizer (Kimble) and solubilized in buffer comprising 20 mM HEPES pH 7.5, 500 mM NaCl, 1 mM EDTA, 1% LMNG, 0.1% CHS, 20% glycerol, 1 mg/ml iodoacetamide, and protease inhibitor cocktail (Roche) for 1 h at 4 °C. The solubilized membranes were isolated by ultracentrifugation with a Ti45 rotor (Beckman) at 138,001 × g for 1 h at 4 °C, followed by incubation with anti-Flag affinity G1 resin (GenScript) for 1 h at 4 °C. The resin was washed in batch with 20 mM HEPES pH 7.5, 300 mM NaCl, 10% glycerol, 1 mM EDTA, 0.1% LMNG and 0.01% CHS. The resin was then transferred to an EconoPac column and further washed with 20 mM HEPES pH 7.5, 200 mM NaCl, 10% glycerol, 1 mM EDTA, 0.01% LMNG, and 0.001% CHS. Proteins were eluted using 20 mM HEPES pH 7.5, 200 mM NaCl, 10% glycerol, 5 mM EDTA, 0.01% LMNG, 0.001% CHS, and 0.4 mg/ml Flag peptide. Eluted proteins were concentrated using an Amicon Ultracentrifugal device (100 kDa cut-off; Millipore) and injected into a Superose 6 10/300 column equilibrated with buffer comprising 20 mM HEPES pH 7.5, 200 mM NaCl, 1.5% glycerol, 5 mM EDTA, 0.01% LMNG, and 0.001% CHS. Eluted fractions containing GPR158, RGS7, and Gβ5 were pooled and concentrated to 4.5 mg/ml using a Vivaspin device (100 kDa cut-off; GE Healthcare) for cryo-EM analysis.

**Cryo-EM sample preparation and data collection for apo GPR158.** Apo GPR158 was freshly prepared for cryo-EM grids immediately after size exclusion chromatography, which was performed on a Superose 6 10/300 column (GE Healthcare) equilibrated with 20 mM HEPES pH 7.5, 200 mM NaCl, 0.01% LMNG, 0.001% CHS, and 5 mM EDTA. Peak fractions containing GPR158 were pooled and concentrated in a Vivaspin device (100 kDa cut-off; GE Healthcare). The concentrated protein was supplemented with 0.1% OG (Anatrace) and incubated for 1–1.5 h prior to vitrification. Samples (3 μl) at a concentration of 12.7 mg/ml were applied to glow-discharged holey carbon grids (C-flat 1.2/1.3 Au 400-mesh grid; EMS). Grids were plunge-frozen in liquid ethane using a Vitrobot Mark IV (Thermo Fisher Scientific) with a blot force of 3 and blot time for 5 s at 100% humidity and 4 °C.

Images were acquired using a Talos Arctica electron microscope (FEI) operated at 200 kV and equipped with a Gatan K3 summit direct electron detector in counting mode (Photon Science Center at Pohang University of Science and Technology) at a nominal magnification of ×100,000. Movies were collected comprising 10,361 micrographs. Datasets were collected with a pixel size of 0.83 Å and a defocus of −0.6 to −1.2 μm. Micrographs were dose-fractionated over 50 frames with a dose rate of 9 electrons per pixel per second and a total exposure time of 3.9 s, resulting in an accumulated dose of 50 electrons per Å².

**Cryo-EM sample preparation and data collection for the GPR158–RGS7–Gβ5 complex.** A total of 3 μl of the GPR158–RGS7–Gβ5 complex was treated with 0.1% OG and applied to glow-discharged holey carbon grids (C-flat 1.2/1.3 Cu 400-mesh grid; EMS). The grids were plunge-frozen in liquid ethane using a Vitrobot Mark IV (Thermo Fisher Scientific) with a blot force of 3 and blot time of 5 s at 100% humidity and 4 °C.

Five datasets were collected using a Talos Arctica electron microscope (FEI) operated at 200 kV and equipped with a Gatan K3 summit direct electron detector in counting mode (Photon Science Center at Pohang University of Science and Technology) at a nominal magnification of ×79,000, corresponding to 1.07 Å/pixel. For the first dataset, 2400 movies were collected with a dose rate of 9.42 electrons per pixel per second for 6.07 s, corresponding to a total dose of 50 electrons per Å². For the second dataset, 2945 movies were collected with a dose rate of 10.422 electrons per pixel per second for 5.47 s, corresponding to a total dose of 50 electrons per Å². For the third dataset, 6497 movies were collected with a dose rate of 14.031 electrons per pixel per second for 4.08 s, corresponding to a total dose of 50 electrons per Å². For the fourth dataset, 5770 movies were collected with a dose rate of 10.077 electrons per pixel per second for 5.68 s, corresponding to a total dose of 50 electrons per Å². For the fifth dataset, 3,001 movies were collected with a dose rate of 17.762 electrons per pixel per second for 3.22 s, corresponding to a total dose of 50 electrons per Å². All datasets were collected using a defocus range of −1.0 to −2.0 μm over 50 frames.

**Data processing for apo GPR158.** Movie frames were aligned using MotionCor2 (ref. [58]), and motion-corrected sums were calculated to estimate the contrast transfer function (CTF) using CTFFIND4 (ref. [59]). Micrographs at low estimated resolution were removed, resulting in 8901 micrographs for data processing. A total of 5,152,593 particles were automatically picked in RELION-3 (ref. [60]) and exported to CryoSPARC v2.15 (ref. [61]). Particles were extracted with a box size of 310 pixels and subjected to several rounds of two-dimensional (2D) classification to eliminate contaminants and false-positive particles. Well-defined 2D classes were selected, resulting in 1,116,476 particles for further processing. Ab initio

reconstruction and heterogeneous refinement in CryoSPARC v2.15 divided the particles into six classes. A dominant class (48.6 %) containing 543,174 particles showed a clear shape for both extracellular and TM domains, and was exported to RELION-3. The exported particles were improved via Bayesian polishing and CTF refinement, followed by another round of 3D classification without alignment. The final dataset of 425,819 particles (78.4%) was subjected to 3D refinement and postprocessing without symmetry, which yielded a map with a global resolution of 3.52 Å according to a FSC criterion of 0.143. A divided class with 180,744 particles in cryoSPARC v2.15 revealed a shorter conformation with a more compact middle linker. The particles were exported to RELION-3 and subjected to subsequent 3D refinement and postprocessing without symmetry, which resulted in a resolution of 5.47 Å. The final maps were sharpened using negative B-factors automatically determined by RELION-3.

**Data processing for the GPR158–RGS7–Gβ5 complex.** For the GPR158–RGS7–Gβ5 complexes, movies from five datasets were aligned and dose-weighted using MotionCor2 (ref. [58]), and CTF parameters were calculated using CTFFIND4 (ref. [59]). After auto-picking the particles using template picker, 1,022,357 particles from 2349 micrographs (dataset 1), 1,147,777 particles from 2853 micrographs (dataset 2), 2,919,355 particles from 6221 micrographs (dataset 3), 1,347,653 particles from 4252 micrographs (dataset 4), and 873,753 particles from 2466 micrographs (dataset 5) were extracted individually using CryoSPARC v3.1 (ref. [61]). The combined particles from all datasets were subjected to several rounds of 2D classification using CryoSPARC v3.1. After exclusion of poorly defined classes, 1,971,816 particles were subjected to ab initio reconstruction to produce an initial 3D model using cryoSPARC v3.1. After heterogeneous refinement, 746,766 particles of one class showing the complex with RGS7–Gβ5 were selected for further ab initio reconstruction using cryoSPARC v3.1. After performing heterogeneous refinement, 411,864 particles for the 2GPR158–RGS7–Gβ5 complex were selected and subjected to two rounds of heterogeneous refinement. In total, 264,464 particles for the 2:1:1 complex were subjected to non-uniform refinement, local motion correction, and global CTF refinement. The processed particles were subjected to global non-uniform refinement yielding a map with a global resolution of 4.31 Å. The resulting map was subjected to particle subtraction to exclude the extracellular domain signal, followed by local non-uniform refinement using a mask on 7TM–RGS7–Gβ5, generating a 4.3 Å map. For the 2:2:2 complex, 375,895 particles with a 2:2:2 complex feature from each heterogeneous refinement were merged and subjected to local motion correction followed by heterogeneous refinement. This generated a map with a global resolution of 4.68 Å. To improve the map for two RGS7–Gβ5 heterodimers and the cytoplasmic helices, particle subtraction was performed to exclude the density for the receptor and detergent micelle. The resulting particles including residual density were subjected to local non-uniform refinement using a mask for two RGS7–Gβ5 heterodimers and the cytoplasmic helices, which produced a map with a global resolution of 4.61 Å. Additionally, to understand dynamics of RGS7–Gβ5 in the 2:2:2 complex, the 3D variability analysis [62] in cryoSPARC v3.1 was performed using the 375,895 particles generated from non-uniform refinement: the calculation was focused on density including the cytoplasmic coiled-coil and two RGS7–Gβ5 complexes with a filter resolution of 9 Å. A movie showing conformational variability was generated using UCSF Chimera v1.15 (ref. [63]) (Supplementary Movie 1).

**Model building.** The atomic model of apo GPR158 was built from the 3.52 Å EM map. Main chain connectivity and secondary structural features were clearly resolved, and we placed α-helices, β-strands, and linking loops of both extracellular and TM domains using COOT [64]. The map of the CR, TM regions, and dimeric interface of the extracellular domain was of high quality, which allowed us to assign all side chains on the placed model. Assignment of side chains was guided by small residues (Gly, Ser, and Val) and bulky residues (Tyr, Phe, and Trp). No clear density was observed for side chains of the PAS domain except for helices α4 and α5; hence, these regions were built as poly-Ala chains. The model was subjected to real-space refinement using PHENIX 1.15.2 with geometry and secondary structural restraints [65,66]. The refined model has a MolProbity [67] score of 1.74 and a clash-score of 3.54 (Supplementary Table 1).

The reported structure of *Bos taurus* RGS7-*Mus musculus* Gβ5 (PDB 6N9G) [47] was used as an initial template to build the 2GPR158–RGS7–Gβ5 model. Three residues differ between the human proteins and each of the RGS7 and Gβ5 proteins from the initial template, and these residues were substituted to the equivalent human residues using COOT; Val291Leu, Ala298Leu and Arg375Lys in RGS7, and Asn34Ser, Asp46Glu and Val284Ile in Gβ5. The structures of human RGS7–Gβ5 and apo GPR158 were placed into the 4.3 Å EM map using the fit-in-map tool of UCSF Chimera v1.14 (ref. [63]). Two turns of TM3 and ICL2 disordered in apo GPR158 were well-defined in the EM map of the complex, into which we manually built a model using COOT. The Ha and Hb helices were built as poly-Ala chains based on the EM maps. Residue numbers were assigned based on a recent publication [7,8], 3D structure prediction using the I-TASSER server [44], and secondary structure prediction using PSIPRED v4.0 (ref. [68]) and XtalPred RF [69]. The 2:1:1 model was subjected to real-space refinement using PHENIX 1.15.2 with geometry and secondary structural restraints [65,66]. The refined model exhibited a MolProbity [67] score of 2.13 and a clash-score of 9.65 (Supplementary Table 1). The following regions are not present in the map and not modeled: residues 1–17 (DEP), 219–255 (GGL), 450–495 (RGS) of

RGS7, and residues 1–14 and 354–395 of Gβ5. For the 2:2:2 complex, we docked the structures of GPR158 and human RGS7–Gβ5 into the cryo-EM map using the fit-in-map tool of UCSF Chimera[63] (Supplementary Fig. 9i, p).

**Adenylate cyclase activation cell-based assay**. The adenylate cyclase activation assay was performed following the procedure from the previous study[11]. GBR1-GFP, GBR2-Flag, GPR158-Flag, RGS7-Flag, and Gβ5 were cloned into the pEG BacMam vector (a gift from Dr. Eric Gouaux). All GPR158 mutations in this study were introduced by PCR-based site-directed mutagenesis and verified by DNA sequencing. Primer sequences are provided in Supplementary Table 2. HEK293T cells were seeded at a density of $0.5 \times 10^6$ cells per well in a six-well plate containing complete growth medium, which consisted of high glucose DMEM (Lonza) and 10% fetal bovine serum (FBS; Welgene), on the day before transfection. HEK293T cells were transfected with 5 μg plasmids encoding GBR1-GFP, GBR2-Flag, RGS7-Flag, Gβ5, GPR158-Flag, and pGloSensor-22F cAMP (Promega) using Fugene HD (Promega). Transfected cells were treated with 1 μM GABA (Tocris Bioscience) for 24 h. Cells were detached with Cell Dissociation Buffer (Gibco) and resuspended in equilibration medium, which comprised 86% $CO_2$-independent medium (Invitrogen), 10% FBS, 4% GloSensor cAMP Reagent solution (Promega), and 1 μM GABA. Cells in the equilibration medium were seeded in a tissue culture treated solid white 96-well plate at a density of 100,000 cells per well and incubated for 2.5 h at room temperature. Expression levels of GBR1-GFP, GBR2-Flag, GPR158-Flag, and RGS7-Flag were examined by western blot analyses. Bioluminescence in the presence of 1 μM GABA was initially quantified using a Centro XS[3] LB 960 microplate luminometer (Berthold Technologies) with an integration time of 1 s. After addition of 250 μM CGP54626 (Tocris Bioscience) in $CO_2$-independent medium, luminescence was subsequently measured every 90 s. The results were normalized to fold changes between values obtained in the absence and presence of CGP54626. Statistical analysis was performed using GraphPad Prism 9.1.1. A one-way analysis of variance (ANOVA) and a multiple comparison correction of Tukey's post hoc test were used to analyze significant differences among conditions.

**Fluorescent size exclusion chromatography analysis**. Oligomeric state of the dimeric interface-disrupting mutant (F135A/F540A/W578A) was analyzed using fluorescent size exclusion chromatography. Genes encoding wild type or mutant full-length GPR158 were fused to GFP and inserted into pEG BacMam vector. The resulting plasmids were transfected into HEK293S GnTI[−] cells which were cultured using the protocol of Goehring et al.[57]. The cells were harvested and solubilized in the buffer containing 20 mM HEPES pH 7.5, 300 mM NaCl, 1 mM EDTA, 2% OG, and 0.2% CHS. After incubating for 1 h at 4 ℃, the solubilized membranes were cleared using ultracentrifugation at $99,278 \times g$ with a TLA45 rotor (Beckman) for 1 h. The supernatants were loaded onto a Superose 6 10/300 column equilibrated with buffer comprising 20 mM HEPES pH 7.5, 200 mM NaCl, 1 mM EDTA, and 0.05% DDM. The eluent was detected by a fluorometer with excitation of 488 nm and emission of 512 nm.

**Subcellular fractionation**. Cells transfected for the adenylate cyclase activation cell-based assay were subjected to the subcellular fractionation to compare expression levels of total, membrane, and cytosolic fractions of GBR1, GBR2, GPR158, and RGS7. The subcellular fractionation was carried out using the procedure from the previous study with minor modifications[7]. Cells were lysed in lysis buffer comprising 50 mM Tris-HCl pH 7.4, 150 mM NaCl, 1 mM EDTA, 2.5 mM $MgCl_2$, and protease inhibitor cocktail (Roche) by sonication. Equal amounts of lysates were ultracentrifuged at $98,384 \times g$ with a TLA-55 rotor (Beckman) for 30 min at 4 ℃. The supernatant containing the cytosolic fraction was transferred to a fresh tube. The pellet was thoroughly washed with the lysis buffer and then incubated in 50 mM Tris-HCl pH 7.4, 300 mM NaCl, 1% Triton X-100, and protease inhibitor cocktail (Roche) for 40 min at 4 ℃. The membrane fraction was isolated by ultracentrifugation at $98,384 \times g$ for 15 min. The lysates in the lysis buffer were supplemented with 1% Triton X-100 and incubated on ice for 1 h. The total fraction was cleared by ultracentrifugation at $98,384 \times g$ for 30 min. For western blot analyses, each fraction sample was resolved by 12% SDS-PAGE and transferred to an Immobilon-P PVDF membrane (Sigma). The following antibodies were diluted to the manufacturers' working concentration and incubated with the membranes for 14 h at 4 ℃; anti-OctA antibody (Santa Cruz Biotechnology, #sc-166355, 1:400 dilution), anti-GFP antibody (Santa Cruz Biotechnology, #sc-9996, 1:40 dilution), anti-glyceraldehyde-3-phosphate dehydrogenase antibody (GAPDH; Sigma, #MAB374, 1:5000 dilution) and anti-N-cadherin antibody (Sigma, #C3865, 1:50). Anti-OctA antibody was used to detect Flag tags from GBR2, GPR158 and RGS7, and anti-GFP antibody was for GBR1. After incubation with anti-mouse IgG secondary antibodies (Santa Cruz Biotechnology, #c-516102, 1:2000 dilution) conjugated with horseradish peroxidase at room temperature for 2 h, signals were developed using SuperSignal West Femto Maximum Sensitivity Substrate (Thermo Fisher Scientific) and detected with a Amersham Imager 680 chemiluminescence imaging system (GE Healthcare). Statistical analysis was performed using GraphPad Prism 9.1.1. A one-way analysis of variance (ANOVA) test and a multiple comparison correction of Tukey's post hoc test were used to analyze significant differences among conditions. Uncropped and unprocessed scan blots are included in the Source data file

**Reporting summary**. Further information on research design is available in the Nature Research Reporting Summary linked to this article.

## Data availability
Atomic coordinates and the cryo-EM map have been deposited in the PDB and the EM Data Bank, respectively, under following accession numbers: EMD-31351 and 7EWL (apo GPR158), EMD-31360 (overall refined 2GPR158–RGS7–Gβ5), EMD-31365 (locally refined 2GPR158–RGS7–Gβ5), and 7EWP (2GPR158–RGS7–Gβ5), and EMD-31363 (overall refined 2GPR158-2RGS7-2Gβ5), EMD-31366 (locally refined 2GPR158-2RGS7-2Gβ5), and 7EWR (2GPR158-2RGS7-2Gβ5). The reported structural model of *Bos taurus* RGS7-*Mus musculus* Gβ5 (PDB 6N9G)[47] was used as an initial template to build the 2GPR158–RGS7–Gβ5 model. Source data are provided with this paper.

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

## Acknowledgements

We thank J.S. Ko for assistance in cloning, and Y.J. Kim and members of the Cho's lab for comments. This work was supported by grants from the National Research Foundation of Korea (NRF) funded by the Korea government (MEST, No. 2021R1A2C301335711 and 2017M3A9F6029736), Samsung Science and Technology Foundation (SSTF-BA1602-14), BK21 program (Ministry of Education) to Y.C. E.J. and Y.K. acknowledge financial support from POSTECH Basic Science Research Institute Grant (2021R1A6A1A10042944).

## Author contributions

E.J. and Y.K. carried out protein expression, purification, and structure determination; E.J., Y.K. and J.J. participated in biochemical experiments; E.J, Y.K., J.J. and Y.C. designed research; and E.J, Y.K. and Y.C. wrote the manuscript.

## Competing interests

The authors declare no competing interests.
