## [Peer Review File · Nature Communications]

Structure of the class C orphan GPCR GPR158 in complex with
RGS7-G β 5REVIEWER COMMENTS

Reviewer #1 (Remarks to the Author):

The manuscript “Structure of the class C orphan GPCR GPR158 in complex with RGS7-Gβ5” by Jeong and colleagues is well written and covers a relevant topic. The authors describe the cryo-EM structure of the orphan G protein coupled receptor GPR158 in complex with the obligatory heterodimer RGS7-Gβ5. If structures of GPCRs are often a source of key information regarding signaling properties of this important class of membrane receptors, in this case, the solved structure could be even more relevant because: 1. This study is focused on an orphan receptor that lacks of pharmacological tools in spite of its known pathophysiological relevance; 2. It shows the rare complex formation between a GPCR and a RGS protein shedding light on this unusual configuration of signaling molecules; and 3. The identification of a PAS domain within the ectodomain of GPR158 will guide future studies towards the deorphanization of this class of orphan receptors (GPR158 and GPR179).

Comments:

- Lines 62-64: measurements of GPR158 constitutive activity indicates a coupling to Gi/o proteins but not Gq (Watkins, 2021 Br J Pharmacol). I would suggest to indicate that the G protein coupling, if any, still needs further investigation especially in the absence of a proven ligand, since no functional assay using osteocalcin as an agonist has ever been reported.
- Fig. 5e. A cell-based assay was performed to measure cAMP levels using a Promega GloSensor. A formal statistical analysis of the different effects of GPR158 mutants is missing. In the text (line 254-255) is reported: “Next, we mutated intra-TM ion-pairs in layer III: E609H (Fig. 3b). The mutation reduced the activity by 50% compared with that of wild-type GPR158 (Fig. 5e)”. This statement requires a proper statistical analysis to support the conclusion that the mutants have any actual effects. In fact, it is unclear where the 50% reduction is. Moreover, it should be verified that co-transfection with GPR158-RGS7-Gβ5 does not affect GABAB expression and membrane localization leading to the observed effects. Again, a statistical comparison showing any significant effect due to the expression of GPR158-RGS7-Gβ5 should be reported. Finally, it was previously reported that overexpression of RGS7-Gβ5 complex leads to the same effect observed using the same assay, therefore it is possible that the reduction of cAMP production observed when transfecting GPR158-RGS7-Gβ5 is only due to overexpression of RGS7. Overall, appropriate controls and statistical analysis are needed to draw any sort of conclusion from this experiment.
- Line 284-285: This sentence is confusing: “Thus, each GPCR must recognize the RGS proteins with high specificity”. RGS proteins recognize specific subsets of Gα proteins that are the substrate of their catalytic GAP activity (Masuho, 2021 Cell). GPCRs activate G proteins, while direct interactions between GPCRs and RGS proteins are rare. In fact, GPR158 represents an exception as it interacts with RGS7 in a very specific way and recruits it to the plasma membrane potentiating its function. I would suggest a major rephrasing of this entire paragraph to improve clarity.
- Line 308-310: I believe the authors are referring to the 3 PDEγ-like (PGL) domains previously identified in the C terminus of GPR158 (Orlandi, 2015 JBC). It was reported that the PGL domains interact specifically with Gai/o-GTP and not with intermediate state or Gai/o-GDP. The purified CT of GPR158 was shown to accelerate the GTP hydrolysis of Gao in solution when added on top of RGS7. However, it was also shown that the effect was dependent on the first half of the CT that only included the RGS7/Gβ5 binding site, but not PGLs. Therefore this data does not support a role for PGLs in the potentiation of RGS7 by GPR158 as suggested.

Minor comments:

- Lines 31-34: the binding site for RGS7-Gβ5 on GPR158 is not clearly defined in the abstract, or at least it does not reflect what reported in the manuscript section “Structures of the GPR158-RGS7-Gβ5 complex”. I would suggest to rephrase this paragraph to improve clarity.
- When describing the ratio between components of the complex, it would help clarity if RGS7 and Gβ5 would be considered separately. I would suggest to describe the complex GPR158-RGS7-Gβ5 as a 2:1:1 ratio and 2:2:2 ratio (for example: lines 25 and 195-196).
- G protein-coupled receptor (GPCR) is defined twice in the introduction (lines 43 and 54).

- When using the terminology “ligand-binding site” referring to Class C GPCRs (i.e. line 142 or Fig3 panel e) it would be more appropriate to discriminate between orthosteric and allosteric sites.
- Line 202: the coiled-coil region mentioned is not highlighted in the figure (Extended fig. 3), or at least the Ha and Hb helices should be described in the figure legend as components of the coiled-coil region.
- Lines 227-229: the removal of the coiled-coil was shown to abrogate binding of GPR158 to the RGS7 complex, but it was also previously demonstrated that the purified GPR158 region 665-775, that includes the coiled-coil region, was sufficient for the binding to purified RGS7-Gβ5 complex in solution (Orlandi 2015 JBC).

Reviewer #2 (Remarks to the Author):

The manuscript entitled “Structure of the class C orphan GPCR GPR158 in complex with RGS7-GB5” by Jeong and co-workers describes three cryoEM structures of the apo GPR158, a 2:1 RGS7-GB5 complex and a 2:2 RGS7-GB5 complex. The paper attempts to leverage the three structures to try and decipher the selective binding of RGS6/7-GB5 to the orphan GPCR and how it could describe the signaling profile of this receptor. In general, the paper is well written and makes comprehensive comparisons to other class C GPCR structures in order to put the reported structures in context. This reviewer also acknowledges that at the time of reading it appears to be the first of its kind of an RGS7-GB5 bound GPCR structure and is of scientific importance.

This reviewer however has some issues with the quality of the structures in this paper. Firstly, the maps have been sharpened with a deep neural network algorithm (DeepEMHancer) which appears to have led to spurious map densities. For example, in the ecto domain map, the region near S261 has some strange map artifacts that appear seem to look like low resolution alpha-helices but do not appear to be able to sensibly connect to anything in the structure. Also, with respect to the ecto structure there are many regions where the molecular coordinates are a poor fit (an example being A85-A98), there are also many Ramachandran outliers (1.95% according to Dynamrama). This reviewer’s concerns are that parts of this particular map are not overly trustworthy and would like to see either the unfiltered maps or standard post-processed maps.

Also, many of the maps look like they have overfitting artifacts. This reviewer would like to see not just the raw FSC curves but the phase-randomized ones, this would alleviate any concerns that the data was overfitted. It appears that either the artifacts are due to overfitting or possibly due to attempting to align a very structurally heterogeneous particle stack, or possibly due to solvent masks that may be too tight.

On this, the locally refined 2:1 and 2:2 complex maps also appear to be overfitted and the mask used in this refinement included too much of the receptor/detergent micelle, such that anything other than rigid body fitting the RGS7-GB5 inappropriate. This reviewer would like to see these maps recalculated with masks that fully exclude any components of the detergent micelle/receptor. Also, the maps for the 2:1 complex appear to contain heterogeneous populations of particles as there appears to be rough density for a 2:2 complex in the 2:1 data. More efforts to in silico purify the particle populations would potentially be help improve the map quality.

For the apo maps/model, this reviewer also has some concerns. The maps were calculated enforcing C2 cyclic symmetry, but it appears that the model was not. On this, how certain is the C2 symmetry, it would be good to see FSC curves for both a C1 and C2 symmetric 3D-reconstruction. The apo structure also has many modelling issues with 11 Ramachandran outliers and some spurious sidechain rotamers. There are also many parts of the ecto domain (in the full apo-structure) which are really not supported by the calculated density (e.g. G291-G302), perhaps these parts of the structure could be omitted as the density isn’t there to support any accurate modelling. The apo-structure also contains many modelled CHS molecules as well as some octyl-glucoside (OG) molecules where do not appear to be well supported by the calculated density. Also this reviewer is not sure how OG

would make it into the structure if only the grids were pretreated with OG and it wasn't present in the mother liquor applied during vitrification?

A very minor comment is the experimental section where blot force is describing blot time (Pg 17 , line 378)

Finally a few comments were made as to a proposed Ca(II) binding site at D314/D316, and while this is certainly a possibility, there is nothing other than the presence of these residues in a suitable position to back up this claim, and while the region bares similarities to the overall fold of Factor IX, it is equally probable that this could be the binding site for other metals or ligands.

I would recommend revisions before publication.

Matthew J. Belousoff (Monash Institute of Pharmaceutical Sciences)

Response to Reviewer #1's comments

(1) Lines 62-64: measurements of GPR158 constitutive activity indicates a coupling to Gi/o proteins but not Gq (Watkins, 2021 Br J Pharmacol). I would suggest to indicate that the G protein coupling, if any, still needs further investigation especially in the absence of a proven ligand, since no functional assay using osteocalcin as an agonist has ever been reported.

>> Please see line 59-63. We have corrected the sentences.

“Activation of the G protein by GPR158 remains unclear: although OCN binds to a complex containing GPR158 and Gαq and regulates the IP3 production, no functional assay using OCN as an agonist has been reported. By contrast, GPR158 exhibits constitutive activity for Gi/o proteins but not for Gq.”

(2) Fig. 5e. A cell-based assay was performed to measure cAMP levels using a Promega GloSensor. A formal statistical analysis of the different effects of GPR158 mutants is missing. In the text (line 254-255) is reported: “Next, we mutated intra-TM ion-pairs in layer III: E609H (Fig. 3b). The mutation reduced the activity by 50% compared with that of wild-type GPR158 (Fig. 5e)”. This statement requires a proper statistical analysis to support the conclusion that the mutants have any actual effects. In fact, it is unclear where the 50% reduction is. Moreover, it should be verified that co-transfection with GPR158-RGS7-Gβ5 does not affect GABAB expression and membrane localization leading to the observed effects. Again, a statistical comparison showing any significant effect due to the expression of GPR158-RGS7-Gβ5 should be reported. Finally, it was previously reported that overexpression of RGS7-Gβ5 complex leads to the same effect observed using the same assay, therefore it is possible that the reduction of cAMP production observed when transfecting GPR158-RGS7-Gβ5 is only due to overexpression of RGS7. Overall, appropriate controls and statistical analysis are needed to draw any sort of conclusion from this experiment.

>> Please see the revised Fig. 5e, Supplementary Fig. 12 and legend (line 935-945, p59), and line 253-261.

We have re-performed the experiments with appropriate controls and statistical analysis. We also examined the expression level and membrane localization of GPR158, RGS7, GBR1 and GBR2 (Supplementary Fig. 12). Based on these data, we have improved the contents of the cell-based assay more clearly.

(3) Line 284-285: This sentence is confusing: “Thus, each GPCR must recognize the RGS proteins with high specificity”. RGS proteins recognize specific subsets of Gα proteins that are the substrate of their catalytic GAP activity (Masuho, 2021 Cell). GPCRs activate G proteins, while direct interactions between GPCRs and RGS proteins are rare. In fact, GPR158 represents an exception as it interacts with RGS7 in a very specific way and recruits it to the plasma membrane potentiating its function. I would suggest a major rephrasing of this entire paragraph to improve clarity.

>> Please see line 290-296. We have rephrased the sentences to improve the clarity.

(4) Line 308-310: I believe the authors are referring to the 3 PDEγ-like (PGL) domains previously identified in the C terminus of GPR158 (Orlandi, 2015 JBC). It was reported that the PGL domains interact specifically with Gai/o-GTP and not with intermediate state or Gai/o-GDP. The purified CT of GPR158 was shown to accelerate the GTP hydrolysis of Gao in

solution when added on top of RGS7. However, it was also shown that the effect was dependent on the first half of the CT that only included the RGS7/Gβ5 binding site, but not PGLs. Therefore this data does not support a role for PGLs in the potentiation of RGS7 by GPR158 as suggested.

>> Please see line 322-324. We thank to the reviewer. GPR158 recruits various states of Gai/o subunit through the first half of the cytoplasmic domain and places the RGS7 and Gai/o proteins in close proximity. We have corrected the sentence.

Minor comments:

(1) Lines 31-34: the binding site for RGS7-Gβ5 on GPR158 is not clearly defined in the abstract, or at least it does not reflect what reported in the manuscript section “Structures of the GPR158-RGS7-Gβ5 complex”. I would suggest to rephrase this paragraph to improve clarity.

>> Please see line 29-31. We rewrote the sentences to describe the interaction between GPR158 and RGS7 more clearly.

(2) When describing the ratio between components of the complex, it would help clarity if RGS7 and Gβ5 would be considered separately. I would suggest to describe the complex GPR158-RGS7-Gβ5 as a 2:1:1 ratio and 2:2:2 ratio (for example: lines 25 and 195-196).

>> We describe a 2:1:1 ratio and a 2:2:2 ratio of the GPR158: RGS7:Gβ5 complex throughout the text. We also describe the complex as “2GPR158-RGS7-Gβ5” and “2GPR158-2RGS7-2Gβ5”.

(3) G protein-coupled receptor (GPCR) is defined twice in the introduction (lines 43 and 54).

>> We removed the second description in the text.

(4) When using the terminology “ligand-binding site” referring to Class C GPCRs (i.e. line 142 or Fig3 panel e) it would be more appropriate to discriminate between orthosteric and allosteric sites.

>> Please see line 142 and Fig. 3e legend (p32, line 774-775). We have specified to “.. the orthosteric or allosteric binding sites”.

(5) Line 202: the coiled-coil region mentioned is not highlighted in the figure (Extended fig. 3), or at least the Ha and Hb helices should be described in the figure legend as components of the coiled-coil region

>> Please see revised Supplementary Fig. 4d and its legend (p44-45, line 842-843). The Ha and Hb helices are marked.

(6) Lines 227-229: the removal of the coiled-coil was shown to abrogate binding of GPR158 to the RGS7 complex, but it was also previously demonstrated that the purified GPR158 region 665-775, that includes the coiled-coil region, was sufficient for the binding to purified RGS7-Gβ5 complex in solution (Orlandi 2015 JBC).

>> Please see line 226-228 (p11). We changed the sentence to “.. The GPR158 fragment (residues 665-775) containing the coiled-coil is sufficient for the binding to the RGS7-Gβ5 complex...”

Response to Reviewer #2's comments

(1) This reviewer however has some issues with the quality of the structures in this paper. Firstly, the maps have been sharpened with a deep neural network algorithm (DeepEMHancer) which appears to have led to spurious map densities. For example, in the ecto domain map, the region near S261 has some strange map artifacts that appear seem to look like low resolution alpha-helices but do not appear to be able to sensibly connect to anything in the structure. Also, with respect to the ecto structure there are many regions where the molecular coordinates are a poor fit (an example being A85-A98), there are also many Ramachandran outliers (1.95% according to Dynamrama). This reviewer's concerns are that parts of this particular map are not overly trustworthy and would like to see either the unfiltered maps or standard post-processed maps.

>> We have re-processed the data and calculated a map in which we improved our model. To resolve reviewer's concern on a deep neural network algorithm (DeepEMHancer), we used standard post-processed maps without using a deep neural network algorithm.

>> To exclude the possibility of overfitting, we assigned the side-chains in the confident regions, which include dimeric interface in the PAS domain, EGF-likier linker and TM domain. No Ramachandran outliers are present in the final models. We are enclosing the recalculated maps along with the revised manuscript.

(2) Also, many of the maps look like they have overfitting artifacts. This reviewer would like to see not just the raw FSC curves but the phase-randomized ones, this would alleviate any concerns that the data was overfitted. It appears that either the artifacts are due to overfitting or possibly due to attempting to align a very structurally heterogeneous particle stack, or possibly due to solvent masks that may be too tight.

>> We have included the phase-randomized FSC curve for apo GPR158 (Supplementary Fig. 2 j, k). For the complex, we collected additional 8700 images for the complex (~ 300,000 more particles). We found out that the calculation of a map from CryoSPARC generated higher quality map than the map from Relion, and thus used the map from CryoSPARC. We present all five FSC curves (no mask, spherical, loose, tight and corrected) from Gold-standard FSC in CryoSPARC (Supplementary Fig. 9). This improved the EM density maps significantly.

(3) On this, the locally refined 2:1 and 2:2 complex maps also appear to be overfitted and the mask used in this refinement included too much of the receptor/detergent micelle, such that anything other than rigid body fitting the RGS7-GB5 inappropriate. This reviewer would like to see these maps recalculated with masks that fully exclude any components of the detergent micelle/receptor. Also, the maps for the 2:1 complex appear to contain heterogeneous populations of particles as there appears to be rough density for a 2:2 complex in the 2:1 data. More efforts to in silico purify the particle populations would potentially be help improve the map quality.

>> The recalculated 4.3 Å maps (both global and local refined map) showed high quality density in the region of TM and RGS7-Gb5. In the 2:1:1 complex map, the interface between

the GPR158 TM and RGS7 (DHEX domain) is very clear. For 2:1:1, we performed particle subtraction to exclude the GPR158 extracellular domain signal and recalculated the map with masks including residual density. The map excluded only GPR158 extracellular domain showed best quality, whereas for the 2:2:2 complex map, using mask excluding detergent micelle/receptor provided the best map. Also, we performed additional heterogeneous refinement using CryoSPARC to purify the particle populations into two homogeneous classes of 2:1:1 and 2:2:2 complexes. This improved the map quality significantly. We described these in the methods and Supplementary Fig. 8.

(4) For the apo maps/model, this reviewer also has some concerns. The maps were calculated enforcing C2 cyclic symmetry, but it appears that the model was not. On this, how certain is the C2 symmetry, it would be good to see FSC curves for both a C1 and C2 symmetric 3D-reconstruction. The apo structure also has many modelling issues with 11 Ramachandran outliers and some spurious sidechain rotamers. There are also many parts of the ecto domain (in the full apo-structure) which are really not supported by the calculated density (e.g. G291-G302), perhaps these parts of the structure could be omitted as the density isn't there to support any accurate modelling.

>> We calculate the map without enforcing C2 symmetry. This is because in a model from the map without symmetry, the entire protomers differ significantly with rmsd values of 3.3 Å, whereas individual domains (PAS, EGF-like linker, and TM domains) exhibit 0.5 to 1.1 Å (Please see Supplementary Fig. 1h, line 80-84). This suggests that the domains are connected somewhat flexible way, resulting the differences in two protomers. The final model shows no Ramachandran outlier as describe above, and we omitted those unclear regions as suggested by a reviewer.

(5) The apo-structure also contains many modelled CHS molecules as well as some octyl-glucoside (OG) molecules where do not appear to be well supported by the calculated density. Also this reviewer is not sure how OG would make it into the structure if only the grids were pretreated with OG and it wasn't present in the mother liquor applied during vitrification?

>> Please see line 390-392 in the methods "...The concentrated protein was supplemented with 0.1% OG (Anatrace) and incubated for 1–1.5 h prior to vitrification". Furthermore, in the recalculated map, OG molecules are more clearly defined. However, we describe that "At the bottom half of the TM interface, we observed well-defined but unknown densities and assigned as putative octyl-β-D-glucopyranoside (OG) and cholesterol hemisuccinate (CHS), which were added during sample preparation" (Please see line 166-168 and Supplementary Fig. 2g).

(6) A very minor comment is the experimental section where blot force is describing blot time (Pg 17, line 378)

>> Please see line 394 and 407, we have specified "blot time".

(7) Finally a few comments were made as to a proposed Ca(II) binding site at D314/D316, and while this is certainly a possibility, there is nothing other than the presence of these residues in a suitable position to back up this claim, and while the region bares similarities to the overall fold of Factor IX, it is equally probable that this could be the binding site for other metals or ligands.

>> Please see line 111-113 "...the conserved D314 and D316 (A loop) and the main chain oxygen atoms of the elongated B loop correspond to the residues that coordinate a Ca²⁺ ion..."

REVIEWER COMMENTS

Reviewer #1 (Remarks to the Author):

The revised manuscript completely satisfies my concerns.

Reviewer #2 (Remarks to the Author):

I am satisfied that the concerns I raised have been addressed and the paper and structures within have been significantly improved. I would recommend for publication.